# Preparation of TiO_2_ Nanorods@Ni-Foam for Photocatalytic Decomposition of Acetaldehyde—In Situ FTIR Surface Investigation

**DOI:** 10.3390/ma18050986

**Published:** 2025-02-24

**Authors:** Piotr Rychtowski, Bartłomiej Prowans, Piotr Miądlicki, Maciej Trzeciak, Beata Tryba

**Affiliations:** Department of Catalytic and Sorbent Materials Engineering, Faculty of Chemical Technology and Engineering, West Pomeranian University of Technology in Szczecin, Pułaskiego 10, 70-322 Szczecin, Poland; bartlomiej.prowans@zut.edu.pl (B.P.); piotr.miadlicki@zut.edu.pl (P.M.); maciej.trzeciak@zut.edu.pl (M.T.)

**Keywords:** photocatalysis, TiO_2_ nanostructures, acetaldehyde removal, alkalic-hydrothermal synthesis

## Abstract

TNR@Ni-foam structures were prepared by an alkaline hydrothermal method in an autoclave in a strongly alkaline medium (10 M NaOH) at 150 °C with further acid washing (0.1 M HNO_3_) and a second hydrothermal treatment in an autoclave at 180 °C. Two TiO_2_ samples were used for preparation: anatase and P25 of mixed anatase and rutile phases. After the first step of hydrothermal treatment, a layered titanate structure was obtained (Na_2_Ti_3_O_7_). Acid washing caused the substitution of Na^+^ by H^+^ and launched the formation of TNR. After the second hydrothermal treatment at 180 °C, for the optimal quantity of acid used for washing (10 mL per 0.75 g of TiO_2_), titania was crystallized to an anatase phase with small quantities of brookite and rutile. The structures obtained from P25 exhibited more brookite and rutile than those based on the anatase precursor. The morphology of TNR@Ni-foam structures was observed by SEM. The obtained composites were tested for acetaldehyde photodegradation (240 ppm in air) during the continuous flow of gas (5 mL/min) through the reactor coupled with FTIR. The most active samples were those obtained from P25, which had a crystalline structure of TiO_2_ and contained the lowest quantity of residue Na species.

## 1. Introduction

In our previous reports, it was demonstrated that titanium dioxide (TiO_2_) loaded on nickel foam was an excellent material for photocatalytic air purification, especially for the acetaldehyde [1] and ethylene [2] decompositions. In the presence of nickel foam, an enhancement of TiO_2_ photoactivity toward acetaldehyde and ethylene decompositions was observed under both ambient conditions and elevated temperatures [3]. The highest decomposition rate was achieved at 100 °C; under these conditions, the reaction yield on TiO_2_-supported nickel foam was doubled in comparison with TiO_2_ only. The performed photocatalytic tests of acetaldehyde and ethylene decompositions in the presence of some reactive scavengers indicated that superoxide anion radicals were the dominant species contributing to their decompositions. It was stated that these species were greatly generated at the interface of TiO_2_ and nickel foam. Therefore, in the presence of nickel foam, the mineralization of both the acetaldehyde and ethylene was greatly enhanced. In such constructed composite, nickel foam acts as an electron supplier, but also improves the transfer of photogenerated charge carriers. What makes these types of composites unique is that they can be used in thermo-photocatalytic processes, where light and elevated temperature create a synergistic effect of thermocatalysis [4,5]. Moreover, the nickel foam structure is highly porous, which increases the available surface area for TiO_2_ loading. The presence of free spaces in the foam structure allows the fluids to flow through the pores. This additional advantage allows the use of metal foam-based composites such as the photocatalytic filter, which purifies the fluid streams from the pollutants [6].

In the presented studies, TiO_2_ nanorods@Ni-foam were synthesized and tested for acetaldehyde decomposition. TiO_2_ nanorods (TNRs) have some advantages over TiO_2_ nanoparticles, such as a higher available surface for molecule adsorption, increased harvesting of light, and better separation of free charges due to the optimal aspect ratio of reduction to oxidation sites [7]. TiO_2_ of various exposed facets has different properties. Performed calculations of 5f-Ti atoms available for e^–^ trapping indicated that, for anatase crystal (101) exposed facets, they were the highest (5.15191 × 10^18^/m^2^) and were medium for the (100) facet (nanorods) with 3.70300 × 10^18^/m^2^, and zero for the (001) facet. On the contrary, 2f-O atoms available for hole trapping were the highest for the (001) exposed facet of anatase (6.96378 × 1018/m^2^), medium for nanorods (5.55450 × 10^18^/m^2^), and lowest for the (101) facet (5.15191 × 10^18^/m^2^) [8]. Our previous studies [1] revealed that electron trapping and formation of O_2_^–•^ radicals on TiO_2_ was a key factor for acetaldehyde decomposition and mineralization. Additionally, hole-trapping sites in TiO_2_ partly contributed to the acetaldehyde decomposition. Therefore, the TiO_2_ nanorod structure should be a good candidate for acetaldehyde decomposition. Other researchers noticed improved mineralization of acetaldehyde on TiO_2_ nanotubes versus TiO_2_ nanoparticles [9]. Therefore, it was considered that elongation of TiO_2_ toward the (100) facet should be beneficial for gas phase photocatalysis.

There are numerous reports on the preparation of TiO_2_ nanorods (TNRs) via the hydrothermal method [10,11,12]. Most of them utilize substrates, such as expensive titanium isopropoxide (TTIP) [13] or volatile titanium tetrachloride (TiCl_4_) [14], and they also require the use of concentrated acids [15]. Although the TNRs obtained that way are well structured in the context of their morphology and photocatalytic performance, this method is not suitable for the preparation of metal-based composites, since metal supports can not only be damaged but also completely dissolved. In order to successfully fabricate the nickel foam/TNRs composite, it is therefore necessary to use a synthesis without strong acids. Existing at the time of preparation, the methods of nickel foam/TNRs [16] are based on organic substrates. Organic-based syntheses are known to reduce the photocatalytic performance of obtained composites used for certain processes [17,18]. This is due to residual, unreacted substrates that form surface carbon deposits.

According to reported studies by other researchers [19], the alkaline synthesis of TiO_2_ leads to the formation of an intermediate phase of the general formula Na_2-x_H_x_Ti_2_O_5_∙1.8 H_2_O, which, upon the hydrothermal treatment and relevant pH, is transformed to TNRs. In this paper, we utilized a non-organic, two-step hydrothermal synthesis with the use of a strong base solution (10 M NaOH). This method was proved to be suitable for obtaining highly crystalline TNRs [20,21]. We utilized this method to prepare a thin titania layer on nickel foam, which resulted in the formation of highly porous, 3D Ni foam/TNR composites. These composites were then tested in the gas-phase continuous flow photocatalytic reactor coupled with FTIR in situ in the process of acetaldehyde photodegradation at 100 °C. This temperature was applied based on our previous experiments, in which we indicated that it was optimal.

In situ FTIR spectroscopy was used in our study as it is a significant technique in the field of gas-phase photocatalysis. Enabling real-time monitoring of transient intermediates provides essential mechanistic insights into the reactions occurring on photocatalytic materials [22,23]. As demonstrated in the work of Zhang et al. [24], using this technique allowed the monitoring of the mechanism of CO_2_ photoreduction on treated TiO_2_. In the work reported by Subbotina and Barsukov [25], this technique allowed some surface peroxides on the TiO_2_ surface to be identified, which appeared to be the key intermediates during the photocatalytic oxidation of gaseous ethanol.

## 2. Materials and Methods

Two-step hydrothermal synthesis was utilized in the preparation of nickel foam/TiO_2_ nanorods (TNRs). According to the morphological phase diagram of TiO_2_-P25 transformation described by other researchers [21], the selected reaction conditions allow TiO_2_ nanotubes/nanorods to be obtained. In order to determine when this approach is appropriate in the case of other types of TiO_2_, in addition to P25, we used the other one, which exhibited only the anatase phase (KRONOClean^®^7050). The properties of both TiO_2_-based materials are listed in Table 1.

In the first step, 60 mL of 10 M NaOH (POCH Avantor; Gliwice, Poland) was mixed with 1.5 g of either P25 (Evonik; Essen, Germany) or KRONOClean^®^7050 (Kronos International; Leverkusen, Germany) in an ultrasonic bath for 15 min. The mixture was then transferred to a Teflon insert (100 mL) of a stainless-steel autoclave. Nickel foam (Nanografi Nano Technology; Ankara, Turkey) 2 × 2 × 0.1 cm was cleaned by rinsing with acetone (Chempur; Piekary Śląskie, Poland) and deionized water, then placed at an angle of 45 degrees to the autoclave wall, together with NaOH/TiO_2_ suspension. The way it was performed is shown in Figure 1. The autoclave was heat-treated in an oven for 24 h at 150 °C. After cooling down to room temperature naturally, the whole mixture was vacuum filtered under 760 mbar and then rinsed with deionized water until a pH of 7 was reached. The pH was controlled via pH test strips (pH-Fix 7.0–14.0; MAcherey-Nagel; Dueren, Germany). Nickel foam was simply rinsed with deionized water in order to get rid of loosely adhered alkaline TiO_2_ particles. After drying in an oven at 70 °C for 24 h, 0.75 g of obtained powder and nickel foam was mixed with a given amount of 0.1 M HNO_3_ (Chempur; Piekary Śląskie, Poland) for 6 h in order to remove Na^+^ cations. The following amounts of HNO_3_ were applied: 0, 5, 10, and 15 mL. The mixture was then filtered under a vacuum and rinsed until a pH of 7 was reached. Both the obtained mixture and nickel foam were then put into the autoclave for the second hydrothermal treatment at 180 °C for 24 h. The mixture and nickel foam were then rinsed with deionized water and dried in an oven at 70 °C for 24 h. Both the powder and the nickel foam/TNRs obtained were further analyzed.

X-ray diffraction (XRD) measurements of obtained TNRs were performed using a diffractometer (PANanalytical, Almelo, The Netherlands) with a Cu X-ray source, λ = 0.154439 nm. The data were collected in the 2θ range of 10–80°; step size of 0.013. The voltage and current parameters used were 35 kV and 30 mA, respectively. Phases were determined on the basis of the standard diffraction data of JCPDS: 01-071-1168 (anatase), 04-007-6488 (brookite), 01-088-1172 (rutile), and 00-059-0666 (Na_2_Ti_3_O_7_) [28]. The phase composition was determined after previous background determination, K-alpha subtraction, and peak determination. Rietveld analysis was then performed. These operations were carried out using HighScore Plus software (version 3.0.5).

SEM (Scanning Electron Microscope) images were taken using model SU8020 (Hitachi, Tokio, Japan). Images were taken to study the morphology and topography of the TNRs/nickel foam composites. The samples were first attached to a circular platform using carbon adhesive tape. The images were taken at 15 kV and the vacuum level of each chamber was 2 × 10^−7^ Pa. The secondary electron detector SE—Top/Upper/Lower detectors and the SE/BSE or SE(TUL) signal mixing function were used.

X-ray fluorescent (XRF) measurements were performed via Epsilon3 spectrometer (PANanalytical, Almelo, The Netherlands) to determine the leftover percentage of sodium content in the powdered samples of TNRs.

Fourier Transform Infrared Spectroscopy in Attenuated Total Reflectance (FTIR-ATR) mode was used to analyze the prepared powder materials. The averages of 32 measurements of FTIR spectra were taken in a wavenumber range of 4000–400 cm^−1^. The atmospheric air just before the measurement was used as the base spectrum.

The photocatalytic test system for acetaldehyde decomposition is shown in Figure 2. The tests were carried out in an in situ FTIR Nicolet iS50 (Thermo, Waltham, MA, USA) with a photocatalytic reactor (Harrick, Pleasantville, NY, USA) where 5 mL/min of ~240 ppm acetaldehyde in air was continuously fed. The gas mixture flowed through the set of 3 layers of circular TNR-Ni foams irradiated from above by the 365 nm UV LED light source with an optical power of 415 mW and optic fiber diameter of 5 mm (LABIS, Warsaw, Poland). The intensity of incident UV radiation was measured using a photo-radiometer, HD2102.1 (TEST-THERM, Kraków, Poland). The obtained UV intensity value measured on the surface of the reactor cover window equaled 20 W/m^2^. The reactor temperature was set to 100 °C. The concentration of the outlet gas stream was measured by a gas chromatograph (SRI, Torrance, CA, USA) equipped with a flame ionization detector. Analyses were carried out under the following conditions: isothermal oven temperature of 130 °C; detector temperature of 250 °C; automatic sampling loop volume of 2 mL; and a metal capillary column (MXT-1301) of 15 m, with an ID of 0.53 mm and 3.00 μm. Both spectra and chromatograms were collected every 5 min for approximately 2 h.

## 3. Results and Discussion

XRD (X-ray Diffraction) measurements were carried out to determine the phase composition of samples. It should be noted that these measurements were carried out using powdered titania samples. Therefore, the identified phase composition does not reflect exactly the composition of the thin titania layers coating nickel foams. Nevertheless, it provides knowledge about the approximated structure of the titania samples grown on a nickel foam surface. The presented diffractograms of titania samples in Figure 3 and Figure 4 indicated that the flushing of titania samples by nitric acid strongly affected the formation of their crystal structure. Detected phases on diffractograms were anatase, brookite, rutile, and Na_2_Ti_3_O_7_, which were determined on the basis of reference cards No.: 01-071-1168, 04-007-6488, 01-088-1172, and 00-059-0666, respectively. Appendix A provides a detailed summary of the obtained phase compositions, as determined by Rietveld refinement, and the analysis plots for each sample can be found in Appendix A.

The main mechanism that takes place during alkaline syntheses was reported elsewhere [10,21,29,30] and can be summarized by the following reactions:3TiO_2_ + NaOH → Na_2_Ti_3_O_7_ + H_2_O(1)Na_2_Ti_3_O_7_ + 2HCl → H_2_Ti_3_O_7_ + NaCl(2)H_2_Ti_3_O_7_ → 3TiO_2_ + H_2_O(3)

Reaction (1) takes place after the first step of the synthesis. Crystalline TiO_2_ used for synthesis transforms into a layered compound containing Na^+^ in the interplanar spaces. The ion can then be washed off by using acid (2), where H^+^ ions substitute the sodium. Acid washing promotes the formation of NTs, which further curl up to give NRs. The intermediate compound crystallized into anatase, brookite, or rutile in the second hydrothermal synthesis step (3). It was reported that the phase and the morphology of the formed nanostructure are strongly dependent on the acid washing. Moreover, the protonated titanate nanostructures led to the formation of the anatase phase, whereas sodium-rich nanostructures led to the formation of a mixture of titania phases [10].

The obtained results showed that the crystal structure of the formed TiO_2_ depended on the quantity of nitric acid used for synthesis. Titania samples originated from anatase (KRONOClean^®^7050) flushed by a small quantity of nitric acid such as 5 mL (A-TNR-5) or prepared without treatment with acid (A-TNR and A-TNR-0) showed the presence of a Na_2_Ti_3_O_7_ phase. Contrary to that, the samples treated with 10 or 15 mL of HNO_3_ (A-TNR-10 and A-TNR-15) showed the structure of anatase with a minority of brookite and rutile or anatase only (intensive reflex at a 2Ѳ angle of 25.2), respectively (Figure 3). 

In the case of samples originating from the mixture of anatase and rutile (Figure 4), the highest content of the Na_2_Ti_3_O_7_ phase was in the case of samples A/R-TNR and A/R-TNR-0. Flushing by the amount of 5 mL nitric acid (A/R-TNR-5) led to the formation of mostly anatase, but a significant amount of brookite and rutile phases were present too. Some leftover amounts of Na_2_Ti_3_O_7_ were still present. The most crystalline was the sample treated with 10 mL of nitric acid (A/R-TNR-10) showing high-intensity reflex at a 2Ѳ angle of 25.2, and as previously stated, some trace amounts of brookite and rutile phases. Surprisingly, the highest amount of acid (15 mL) used (sample A/R-TNR-15) led to the return formation of the Na_2_Ti_3_O_7_ phase by removing all the TiO_2_ crystalline phases such as anatase, brookite, and rutile. It is worth noting that small intensity reflexes of Ni(OH)_2_ were also detected, similarly as reported elsewhere [31]. The presence of the Ni(OH)_2_ phase in titania powder can be explained by its splintering together with TiO_2_ from the nickel foam. Most probably, Ni(OH)_2_ was formed on nickel foam soaked in a strong alkali solution. The formation of Ni hydroxide on the surface of Ni foam in an alkaline solution (KOH) was described by Kai Wan et al. [32]. They reported that the formation of the NiOOH layer reduces interfacial electronic resistance. Therefore, it is stated that the formation of Ni hydroxide on the Ni foam surface can facilitate electron transfer between TiO_2_ and Ni foam and contribute to the separation of free charges.

The first group of the samples tested were ones that originated from the anatase, untreated and treated with 5 mL of nitric acid (A-TNR-0 and A-TNR-5), presented in Figure 5 and Figure 6, respectively. These composites exhibited similar and relatively even titania coating across the whole surface of regular nanostructures. These titania nanostructures are often referred to as nano-sheets. In addition, larger clusters of titania agglomerates in the form of lumps with a diameter of approximately a few to several micrometers can be observed for the A-TNR-0 and A-TNR-5 samples (Figure 5A vs. Figure 6A).

Samples originated from anatase and treated with 10 and 15 mL of nitric acid (A-TNR-10 and A-TNR-15) are illustrated in Figure 7 and Figure 8, respectively. The surface of both composites was uniformly coated with TiO_2_ nanorods (Figure 7C and Figure 8C). Interestingly, clusters of larger flower-like structures or bouquets with sizes much larger than the nanostructures on the underlying nickel foam itself were also visible (Figure 7A,B; Figure 8B,C). Similar structures were also observed elsewhere [33]. These clusters were significantly larger for sample A-TNR-10. The level of surface coating of A-TNR-15 (Figure 8A,B) allowed the observation of nickel grain borders, whereas, in the case of samples A-TNR-0, A-TNR-5, and A-TNR-10, this was not observed, indicating significantly thicker surface coating in those mentioned samples.

The next composite studied (Figure 9) was the one that originated from the mixture of anatase and rutile and was untreated with nitric acid (A/R-TNR-0). This material was homogeneously covered with two-dimensional nanostructures, often referred to in the literature as nano-sheets or nano-ribbons [10]. Those structures are several nanometers thick, while their length is as large as 2 μm. In some areas, larger agglomerates with dimensions of several to tens of micrometers have been observed (Figure 9A,B). The observed nanostructures were irregularly arranged on the surface and bent or twisted.

The sample treated with 5 mL of nitric acid and originated from the mixture of anatase and rutile (A/R-TNR-5) is illustrated in Figure 10. In this case, TiO_2_ was homogeneously coated on nickel foam and presented elongated particles narrowed at the end, similar to TiO_2_ nanorods. Their average length was around 1.15 μm (Figure 10C). In Figure 10D, vertically oriented nanorods with rectangle shape and side dimensions of a few tens of nm are observed.

In Figure 11, the morphology of P25 treated with 10 mL of nitric acid (A/R-TNR-10) is presented. In this particular case, the nickel foam surface is also coated with TiO_2_ nanorods. However, their dimensions are much larger than that previously observed: their average length was around 2.4 μm (Figure 11B,C). In Figure 11D, we can observe one of the nanorod planes vertically oriented toward the observer with a size reaching a couple of hundred nanometers (Figure 11D). It is worth noting that there are some areas of TiO_2_ agglomerates (Figure 11A). TiO_2_ nanorods, however, accounted for most of Ni foam coverage.

The last sample presented was the one that originated from P25 treated with 15 mL of nitric acid (A/R-TNR-15). This time, TiO_2_ nanorods are in the mixture with some other nanostructures (Figure 12C,D). The obtained, additional structures can be described as star-like nanostructures. What is more, the nickel foam surface was also covered with macro-sized particles (Figure 12A) between which the aforementioned nanostructures are present.

The sodium content was determined via the XRF technique and is presented in Table 2. The leftover sodium was present in the studied TNR samples due to the use of a strong base during synthesis. Interestingly, only samples originating from the mixture of anatase and rutile (A/R-TNRs) indicated significant sodium content. This content was lower the greater the amount of nitric acid used for sample washing. The sodium content was highest when the sample was not washed, which occurred in the A/R-TNR-0 sample. On the other hand, samples originating from the anatase phase (A-TNRs) showed no sodium presence measured via XRF at all. This may be explained by the origin of the sample, which was obtained by the sulfate method, showing on its surface the presence of sulfate groups, which are base centers according to the Lewis theory. Consequently, they may already compete for surface active sites with the strong 10 M NaOH base acting on this type of TiO_2_ at the preparation stage itself. It is therefore highly likely that a simple rinse with deionized water was sufficient in this case to flush out the sodium, which was weakly bound to the formed TNR surface. Samples derived from a mixture of anatase and rutile (A/R-TNR) apparently required harsher treatment (by nitric acid), as the sodium may have embedded itself in the structures of the resulting material, perhaps in the interplanar spaces. This was also observed by other groups [19,34].

FTIR-ATR analysis of the powder samples obtained is presented in Figure 13. Note that they were tested without the presence of nickel foam. One of the noticeable differences between the samples here is in the intensity of the band originating from the hydroxyl groups, located at a wavenumber of 1620 cm^−1^. According to other studies, the transformation of the phases of TNRs during the heating is strongly related to the dehydration process of the titania layered structure [19]. This hypothesis strongly overlaps with our findings, as high hydroxylation of TiO_2_ usually implies incomplete conversion of the titanate phase into anatase, rutile, or brookite. Accordingly, samples such as A-TNR, A/R-TNR-0, A/R-TNR-5, and A/R-TNR-15 exhibited the strongest intensity of the –OH band and were consequently not fully crystallized into TiO_2_. At the same time, samples of the weakest –OH band intensity, such as A-TNR-10, A-TNR-15, and A/R-TNR-10, were the ones that revealed a high-intensity anatase reflex on XRD patterns. The other band, which appeared at 1375 cm^−1^, can be assigned to the vibrations of nitrate anions [35]. An additional band can be observed at 1332 cm^−1^, which is most likely due to the presence of either carbonates or bicarbonates, and is more intensive, the more alkalic the sample surface is [36]. This band disappeared after acid washing with 10 mL and crystallization of TiO_2_. However, under an excess of HNO_3_ washing (15 mL), this band emerges again on the FTIR spectra of both titania samples. NaOH treatment caused disruption of the titania crystal structure and increased the affinity of the titania surface to CO_2_ adsorption. The observed band at around 1550 cm^−1^ can be assigned to carbonate species and that at 1080 cm^−1^ to C-O vibrations. The intensity of these bands decreases with the rinsing of Na.

Photocatalytic tests for acetaldehyde removal are shown in Figure 14. These tests were performed at an elevated temperature (100 °C) to increase the overall process performance due to the catalytic activation of nickel foam. This phenomenon was reported in our previous paper [1]. Because of the relatively low flow rate of gas entering the reactor, which favors adsorption, the experiment could be divided into two stages: adsorption and photocatalytic decomposition. At the low flow rate used, the adsorption step took about 20 min. All the samples except one, originating from P25 untreated by the nitric acid (A/R-TNR-0), showed significant adsorption of acetaldehyde with a maximum of around 220 ppm for A-TNR-10. Although the acetaldehyde adsorption was high on the prepared samples, its photocatalytic decomposition was rather poor, only A/R-TNR-10 and A/R-TNR-15, originating from P25, showed noticeable activity. This was probably caused by the strong adsorption of formed carbonate species on the titania surface upon acetaldehyde oxidation and their resistance to further decomposition.

To investigate exactly what was happening on the surface of individual samples, in situ FTIR studies were performed.

In situ FTIR spectra were recorded directly during the adsorption/photocatalytic process of acetaldehyde removal. The most significant changes were observed in the wavenumber range of 2000–1200 cm^−1^, so it was presented in detail for all spectra together with mappings. Both the appearance and disappearance of infrared bands can be observed here, indicating the loss of given surface groups while the growth of other bands was observed, which is evidence of the formation of intermediate products on the surface of the TNR/nickel foam composites studied. In Figure 15, an in situ FTIR analysis was presented during the aforementioned processes with the use of anatase-originated TNR, untreated by the nitric acid (A-TNR-0). Of all the samples, the largest number of new bands appeared here. As the photocatalytic process progresses, a significant increase in the band intensity at 1690 and 1653 cm^−1^ can be observed. Both can be assigned to ν(C=O) vibrations typical of acetic acid. Moreover, a significant decrease in the δ(OH) 1620 cm^−1^ band was observed, which can be assigned to the presence of surface water, and its reduction over time was due to the elevated temperature of the process. This is the case for all samples. Interestingly, this signal slightly elevates at the last process stage (120 min). Secondly, the increasing band at 1583 cm^−1^ can be assigned to $\nu_{as}(COO)$ and is due to the presence of acetate species. The next bands at 1428, 1417, 1390, and 1379 cm^−1^, which are close to each other, can be assigned to $\nu_{s}(COO)$, $\delta_{s}(CH_{3})$, $\nu_{s}\left(COO\right),$ and $\delta (CH_{3})$, respectively. Their presence is evidence of the appearance of such compounds as acetaldehyde, acetic acid, and crotonaldehyde, with the former being the result of adsorption, and the others formed as intermediate products. Their intensity grows in time. A small intensity band at 1334 cm^−1^ indicates the $\delta_{s}(CH_{3})$ mode and is due to the formation of acetate species. The quite intensely growing band at 1307 cm^−1^ can be assigned to ν(C−O), which results from the presence of either acetic acid or crotonaldehyde. The last band of relatively weak intensity at 1254 cm^−1^ of the ν(C-C) mode comes from the acetic acid [37,38,39].

The next sample studied (Figure 16) originated from anatase and was treated with 5 mL of nitric acid (A-TNR-5), and indicated a much lower number of new bands grown than that observed previously. Surface hydroxyl group reduction is observed at 1620 cm^−1^. Simultaneously with the disappearing of hydroxyl groups, the new bands are observed at 1725 and 1653 cm^−1^, which can be assigned to the stretching vibrations of the C=O group in the adsorbed formic acid and formaldehyde, respectively [40].

Very slight increase in the 1580 cm^−1^ band from the $\nu_{as}(COO)$ mode is observed, which indicates the formation of very low amounts of surface acetate species. The next band at 1418 cm^−1^ of the $\nu_{s}(COO)$ mode results from the significant increase in acetate species throughout the photocatalytic process. The last band of low intensity observed at 1320 cm^−1^ can be assigned to $\delta (CH_{3})$, which comes from some of the organic species adsorbed to the surface [37,38,39].

The sample treated with 10 mL of nitric acid and originating from anatase (Figure 17) exhibited drastically lower band intensities in comparison to different samples. This was most likely due to the much thinner TiO_2_ species layer on the surface of the nickel foam. Nickel is a strongly absorbing material in the infrared range, so the signals were significantly weakened. Once again, the 1620 cm^−1^ band of hydroxyl groups reduced with time. Interestingly, the 1592 cm^−1^ band of $\nu_{as}(CO)$ and the 1361 cm^−1^ band of $\nu_{ss}(CO)$ also reduced with time. Mino et al. [41] proved that these bands show CO_2_ bound to the anatase surface via adsorption and as a form of monodentate carbonates. This may therefore be explained by the desorbing of carbon dioxide from the surface in our case. The band at 1430 cm^−1^ might result from the presence of acetate species and can be assigned to $\nu_{s}(COO)$. Additionally, the band at 1320 cm can be assigned to $\delta \left(CH_{3}\right),$ resulting from the adsorption of acetaldehyde or acetic acid [37,38,39].

Although the sample treated with 15 mL of nitric acid and originating from anatase (Figure 18) was slightly different on the basis of its crystalline structure, its spectra were almost identical to the previously discussed sample. Again, both loss of hydroxyl groups (1620 cm^−1^) and desorbing CO_2_ are observed (1575 and 1353 cm^−1^). A notable band intensity increase over time can be seen at 1313 cm^−1^, which can again be assigned to the $\delta \left(CH_{3}\right)$ of adsorbed organic species [37,38,39].

In the case of the sample originating from the mixture of anatase and rutile, untreated with nitric acid (Figure 19), the spectra changed the most at the very beginning of the process, but then remained the same. A notable increase in band intensity at 1770 cm^−1^ of $\nu_{as}(COO)$ can be observed, which cannot be assigned to only one molecule—its presence most likely resulted from the formation of acetic acid or acetate species. The band at 1652 cm^−1^ can be assigned to CO_2_ bounded with hydroxyl groups [41]. The band at 1620 cm^−1^, still most likely originating from hydroxyl groups attached to the surface, is stretched, and its intensity decreases dramatically. On the other hand, the intensity of the band at 1575 cm^−1^ decreases slightly at the beginning to return to the original level after a while. We therefore assume that this time they are acetate species thanks to the $\nu_{as}(COO)$ mode [37,38,39]. The band at 1369 cm^−1^ of $\nu_{ss}(CO)$ was present at the sample surface and decreased throughout the process, and this is most likely due to the desorption of CO_2_ attached to the surface.

The next sample was the one that originated from the mixture of anatase and rutile and was treated with 5 mL of nitric acid (A/R-TNR-5). Two significant major bands (Figure 20) that decrease over the process duration can be noticed at 1620 and 1589 cm^−1^, which can be assigned to the hydroxyl groups (δ(OH)) and carbon dioxide ($\nu_{as}(CO))$, respectively. A noticeable $\nu_{s}(COO)$ band intensity increase was at 1410 cm^−1^ and was due to the formation of acetate species [37,38,39].

The next sample studied, which was treated with 10 mL of nitric acid and originated from the mixture of anatase and rutile (A/R-TNR-10), exhibited quite a stable surface (Figure 21). A slight decrease in intensity of bands at 1658 and 1620 cm^−1^ of $\nu_{as}(CO$) and δ(OH), respectively, can be noticed. According to the literature [41], the former is derived from CO_2_ bound to surface hydroxyl groups, and the latter is the hydroxyl groups themselves. These signals are therefore related to each other and also decrease together as the process proceeds. Some increasing bands at 1414 cm^−1^ of $\nu_{s}(COO)$ could be noticed, which were most likely due to the formation of surface acetate species [37,38,39]. The last two bands at 1371 and 1355 cm^−1^ were relatively stable over the process. Both bands are once again most likely linked to CO_2_ attachment to the surface, with the former being $\nu_{ss}\left(CO\right),$ and the latter remaining unidentified.

The last sample studied (Figure 22) was the one that originated from the mixture of anatase and rutile and was treated with 15 mL of nitric acid (A/R-TNR-15). Some small decrease in intensity of the 1620 cm^−1^ band of hydroxyl surface groups was observed. Close to it, another band (1588 cm^−1^) can be observed, whose intensity increases over time and can be assigned to $\nu_{as}(COO)$ of either acetate or formate species. Additionally, three time bands increasing over at 1416, 1380, and 1308 cm^−1^ were observed, identified as the $\nu_{s}(COO)$ of acetate species, $\delta (CH_{3})$ and $\delta_{s}(CH_{3})$, respectively. The last two bands were most likely the cause of the crotonaldehyde formed on the surface [37,38,39].

## 4. Conclusions

TNR@Ni-foam structures were prepared by the alkaline hydrothermal method, which consisted of two steps: (1) treatment of TiO_2_ powder in an autoclave in 10 M NaOH at 150 °C for 24 h, then washing with 0.1 M HNO_3_ and neutralization with deionized water; and (2) hydrothermal treatment in an autoclave at 180 °C for 24 h. In the first step of preparation, the layered titanate structure (Na_2_Ti_3_O_7_) was formed. This obtained material was then washed with various quantities of HNO_3_, which strongly affected the formation of TNR in the next step of hydrothermal treatment. It appeared that regardless of the titania precursor used (anatase or mixture of anatase and rutile), for 10 mL of HNO_3_ solution used per 0.75 g of TiO_2_, the highest crystalline structure of TNR was obtained. XRD measurements showed that these TNR structures consisted of anatase with small quantities of brookite and rutile. TNR@Ni-foam structures obtained from the P25 titania precursor contained more brookite and rutile than those prepared from anatase and had a structure of horizontally grown NR on Ni foam. In the case of the anatase precursor used, the hierarchical structures of titania nano-ribbons were observed on the Ni foam. These TNR structures were disrupted by an excess of HNO_3_ used for titanate washing (15 mL per 0.75 g of TiO_2_). XRD measurements showed that the crystallinity of TiO_2_ decreased when an excess of HNO_3_ was used during preparation. Additionally, in the case of the sample obtained from the P25 precursor and the HNO_3_ washing of 15 mL, the Ni(OH)_2_ phase was determined by the XRD pattern. In fact, the role of acid washing is the substitution of Na^+^ to H^+^ in layered Na_2_Ti_3_O_7_ to initiate the process of TNR formation. After acid washing in the second step of hydrothermal treatment, there is a process of titanate dehydration and crystallization. However, when there is an excess of hydrogen ions in the solution, the protonation of TiO_2_ takes place, and there is a possible increase in titania hydration. This effect was observed in the FTIR spectra observed for the prepared powdered titania nanostructures. Interestingly, the obtained TNR@Ni-foam structures from the anatase precursor did not contain any Na species (as indicated by XRF measurements), whereas those prepared from P25 had residue Na, which was lower for the higher quantity of HNO_3_ used for titanate washing. The samples prepared from P25 with a small content of Na indicated higher photocatalytic activity toward acetaldehyde decomposition than those obtained from anatase-type TiO_2_. One of the reasons is related to the phase composition; the presence of the brookite phase in combination with anatase is beneficial for acetaldehyde decomposition [42], and the samples obtained from P25 had a higher quantity of brookite. The other factor affecting the photocatalytic activity of these TNR@Ni-foam structures is related to the thickness of the titania layer and the acidity of the surface. P25 has a low density in comparison with KRONOClean^®^7050 and good dispersion properties, so can be abundantly attached to the Ni foam surface. The acidic surface of TiO_2_ is detrimental to acetaldehyde decomposition [43]; moreover, strong adsorption of acetate species on certain titania centers upon acetaldehyde decomposition can deactivate the photocatalyst when the process is carried out under a continuous flow of reacted gases. It is assumed that TNR@Ni-foam structures prepared from P25 had a more alkaline surface than those obtained from the anatase precursor because of the presence of Na species and, for that reason, could be more active for acetaldehyde decomposition. TNR@Ni-foam structures obtained from the anatase precursor were more hydrophilic and revealed the desorption of water molecules from the surface upon photocatalytic process carried out at 100 °C, as was monitored by FTIR spectra (the band at 1620 cm^−1^). Physically adsorbed water on the titania surface is detrimental to the photocatalytic process carried out in a gas phase because it limits the adsorption of organic pollutants on the titania surface and oxygen, which takes place during the formation of superoxide anion radicals. This was an additional reason for the weak activity of TNR@Ni-foam structures prepared from the anatase precursor. A thin layer of TiO_2_ coating gives the possibility of acetaldehyde adsorption on the nickel foam surface. These adsorbed species do not proceed to photocatalytic decomposition. Generally, P25 appeared to be a good titania precursor for the preparation of TNR@Ni-foam composites, because of the stronger bound sodium species, which contributed to the formation of the layered Na_2_Ti_3_O_7_ structure, which was an intermediate to TNR formulation. Moreover, mixed TiO_2_ phases were crystallized with the dominant anatase phase, which was convenient due to its photocatalytic properties. It was shown that at the proposed conditions of TNR@Ni-foam preparation, Ni(OH)_2_ species were formed, which can contribute to the enhancement of charge carrier separation due to their ability to reduce barriers for electron transfer. Therefore, the A/R-TNR-15 sample revealed high photocatalytic activity, although it had a smaller crystalline structure than A/R-TNR-10, but a thick titania coating.

## Figures and Tables

**Figure 1 materials-18-00986-f001:**
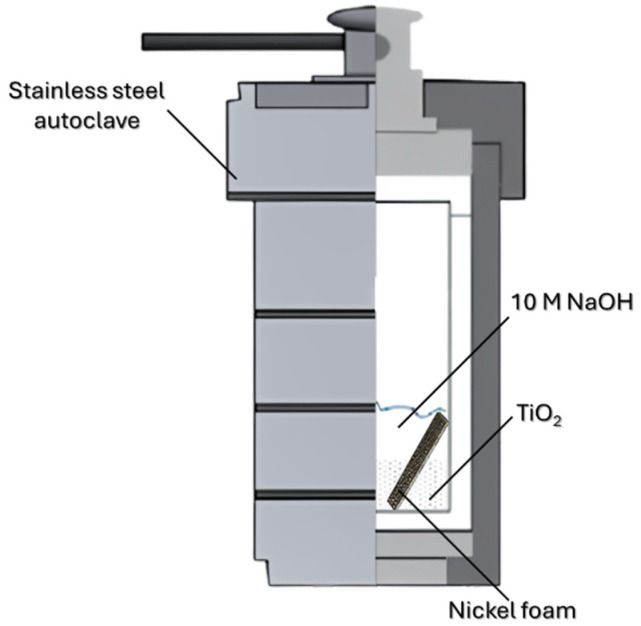
Preparation scheme of TNR-nickel foam composites.

**Figure 2 materials-18-00986-f002:**
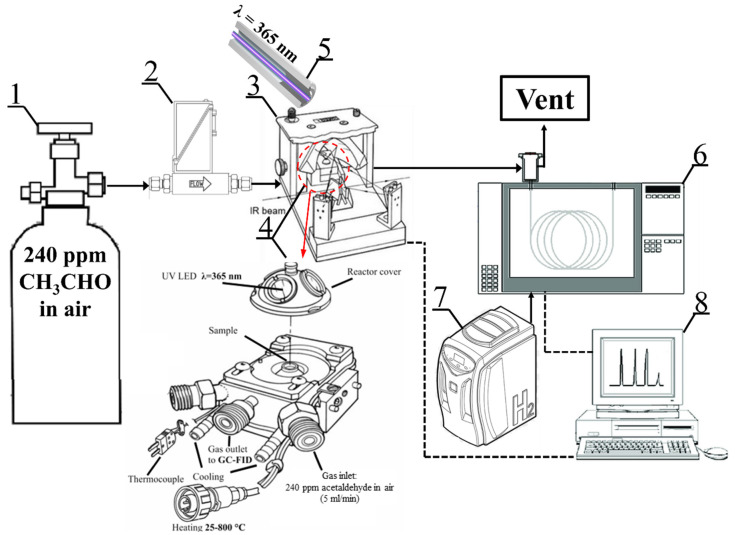
Schematic diagram of photocatalytic reaction system: 1—model gas bottle of 240 acetaldehyde in air, 2—mass flow meter, 3—in situ FTIR Praying Mantis™, 4—high-temperature reaction chamber, 5—optical fiber UV-LED light source, 6—gas chromatograph with FID, 7—hydrogen generator, 8—PC.

**Figure 3 materials-18-00986-f003:**
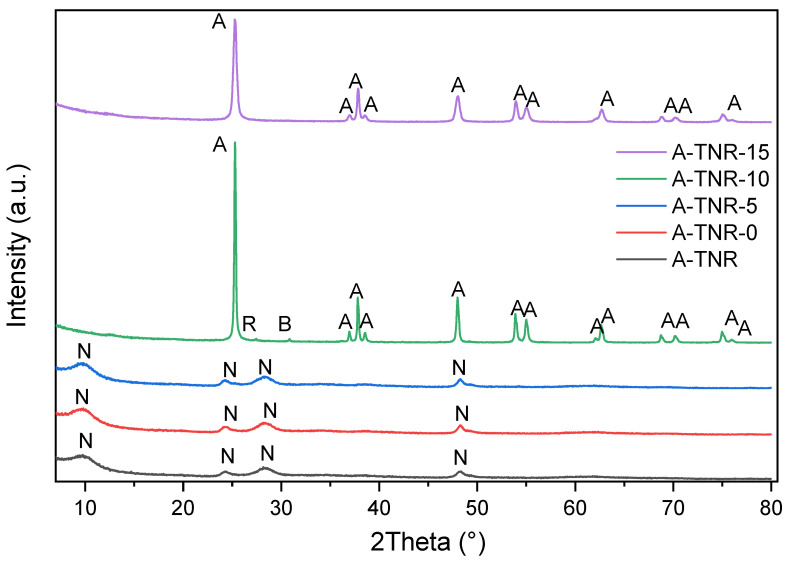
XRD diffractograms of anatase-originated TNRs (without nickel foam). A—anatase, B—brookite, R—rutile, N—Na_2_Ti_3_O_7_.

**Figure 4 materials-18-00986-f004:**
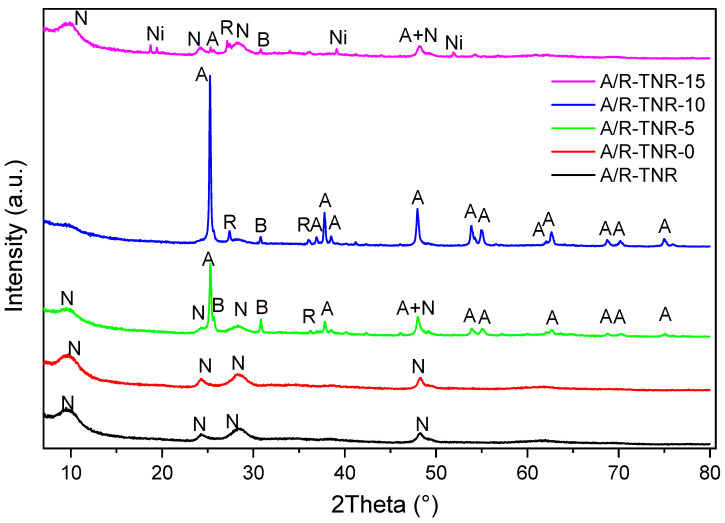
XRD diffractograms of anatase- and rutile-originated TNRs (without nickel foam). A—anatase, B—brookite, R—rutile, N—Na_2_Ti_3_O_7_, Ni—Ni(OH)_2_.

**Figure 5 materials-18-00986-f005:**
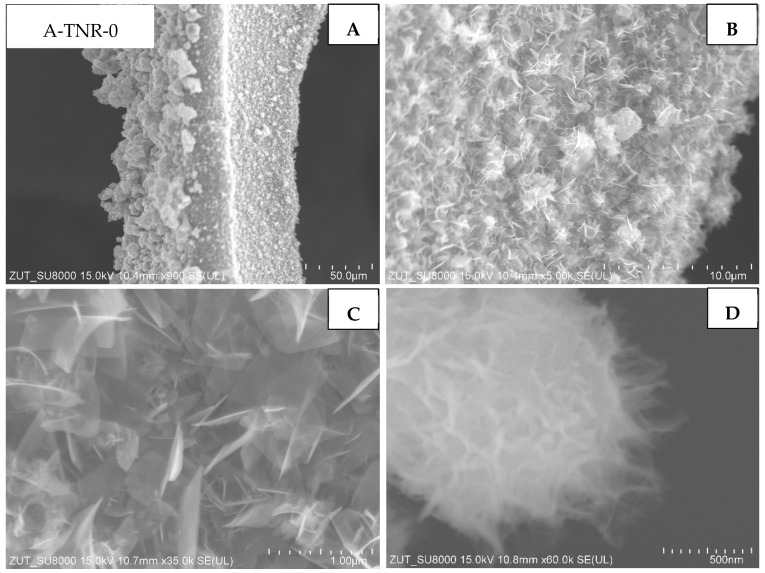
SEM images of A-TNR-0 in magnification: (**A**) 50 µm, (**B**) 10 µm, (**C**) 1 µm, (**D**) 500 nm.

**Figure 6 materials-18-00986-f006:**
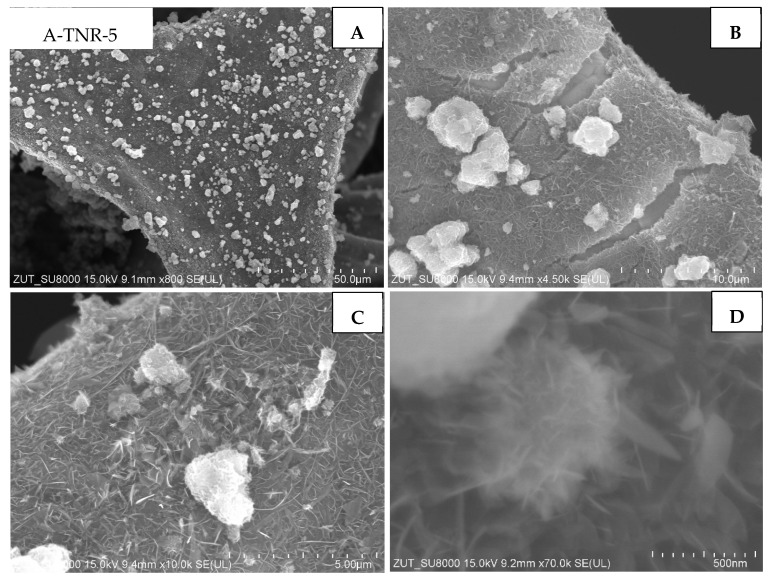
SEM images of A-TNR-5 in magnification: (**A**) 50 µm, (**B**) 10 µm, (**C**) 5 µm, (**D**) 500 nm..

**Figure 7 materials-18-00986-f007:**
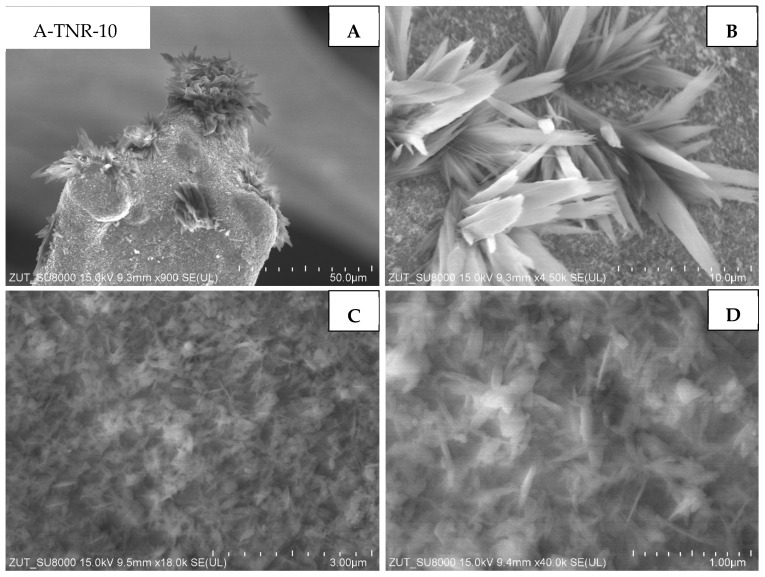
SEM images of A-TNR-10 in magnification: (**A**) 50 µm, (**B**) 10 µm, (**C**) 3 µm, (**D**) 1 µm.

**Figure 8 materials-18-00986-f008:**
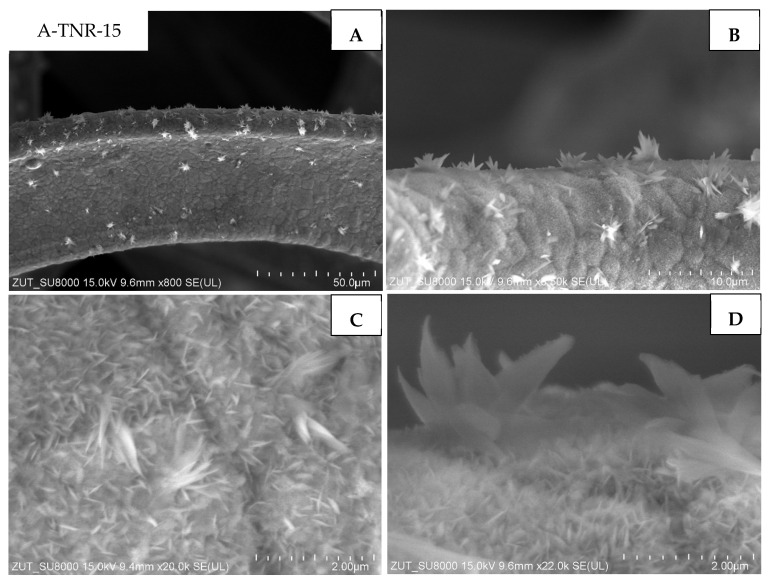
SEM images of A-TNR-15 in magnification: (**A**) 50 µm, (**B**) 10 µm, (**C**,**D**) 2 µm.

**Figure 9 materials-18-00986-f009:**
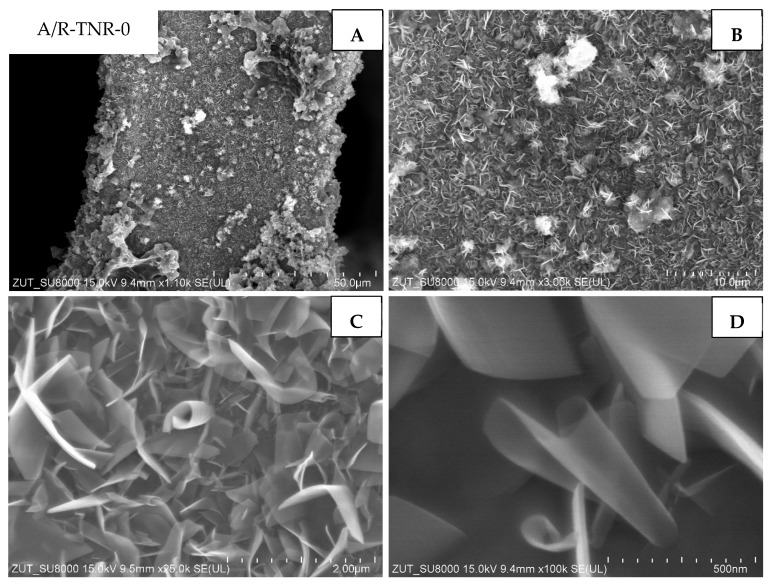
SEM images of A/R-TNR-0 in magnification: (**A**) 50 µm, (**B**) 10 µm, (**C**) 2 µm, (**D**) 500 nm.

**Figure 10 materials-18-00986-f010:**
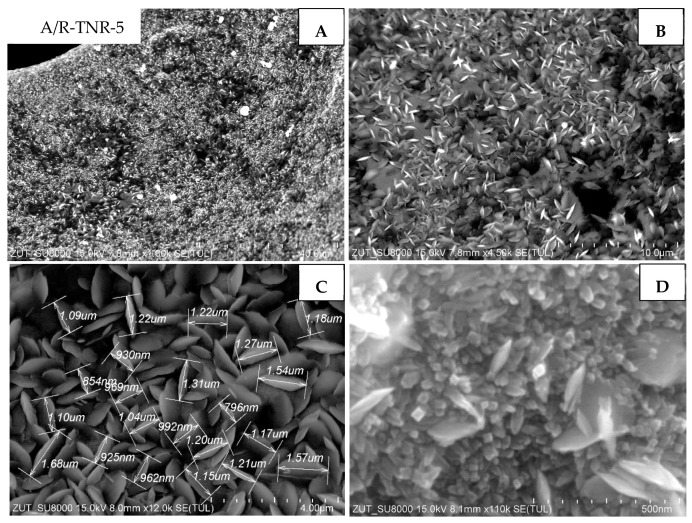
SEM images of A/R-TNR-5 in magnification: (**A**) 50 µm, (**B**) 10 µm, (**C**) 4 µm, (**D**) 500 nm.

**Figure 11 materials-18-00986-f011:**
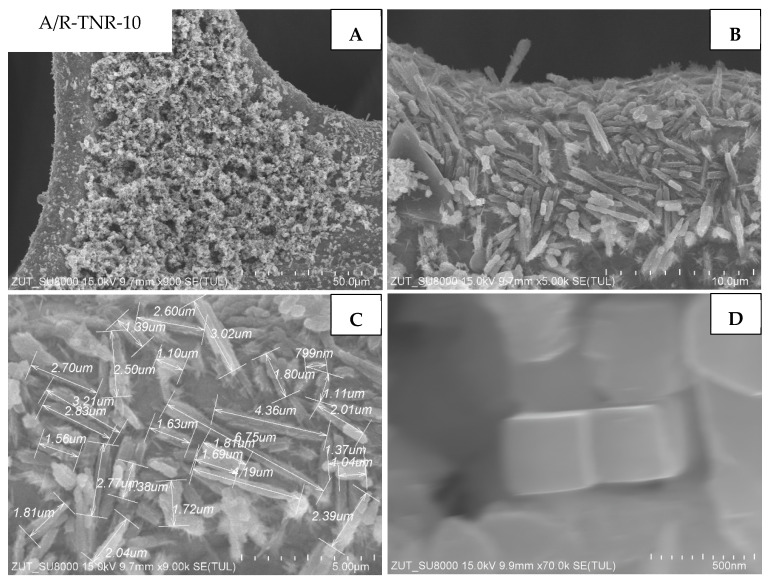
SEM images of A/R-TNR-10 in magnification: (**A**) 50 µm, (**B**) 10 µm, (**C**) 5 µm, (**D**) 500 nm.

**Figure 12 materials-18-00986-f012:**
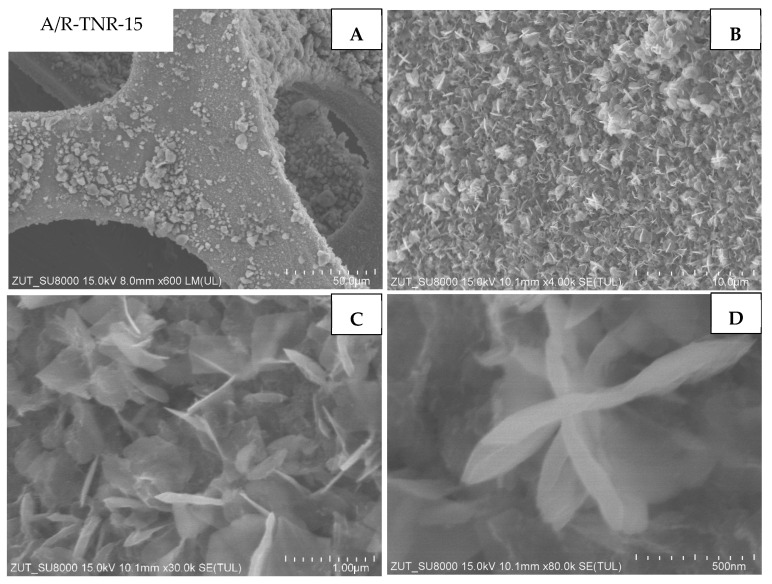
SEM images of A/R-TNR-15 in magnification: (**A**) 50 µm, (**B**) 10 µm, (**C**) 1 µm, (**D**) 500 nm.

**Figure 13 materials-18-00986-f013:**
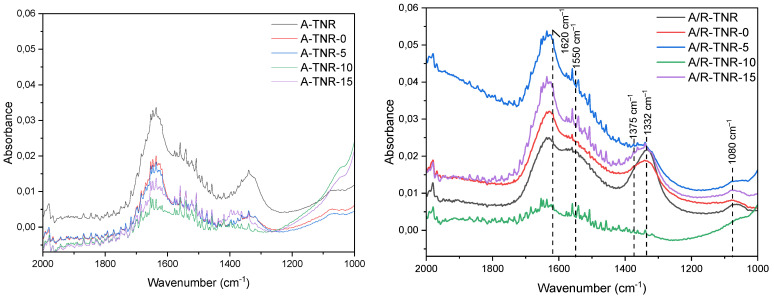
FTIR-ATR analyses of the powder materials obtained.

**Figure 14 materials-18-00986-f014:**
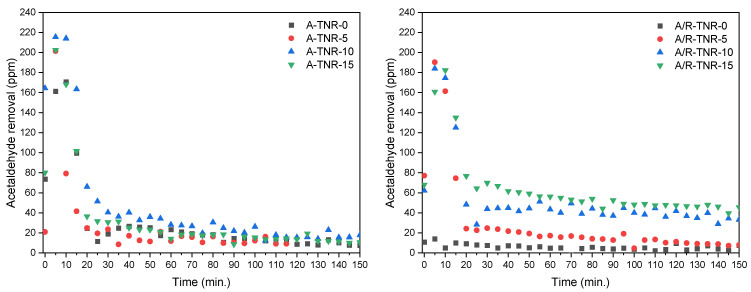
Photocatalytic decomposition of 240 ppm acetaldehyde in air using TNRs/nickel foam composites. Flow rate: 5 mL/min; temperature: 100 °C.

**Figure 15 materials-18-00986-f015:**
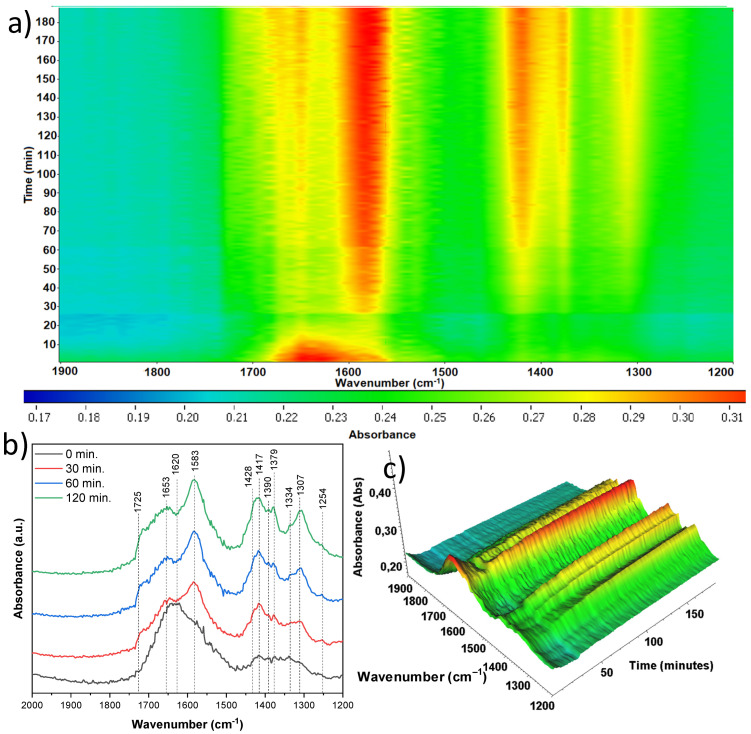
In situ FTIR of A-TNR-0 during the photocatalytic removal of acetaldehyde: (**a**) 2D mapping projection of the A = f(t, ν) relationship, (**b**) A = f(ν) graph after 0, 30, 60, and 120 min, (**c**) 3D mapping of the A = f(t, ν) relationship.

**Figure 16 materials-18-00986-f016:**
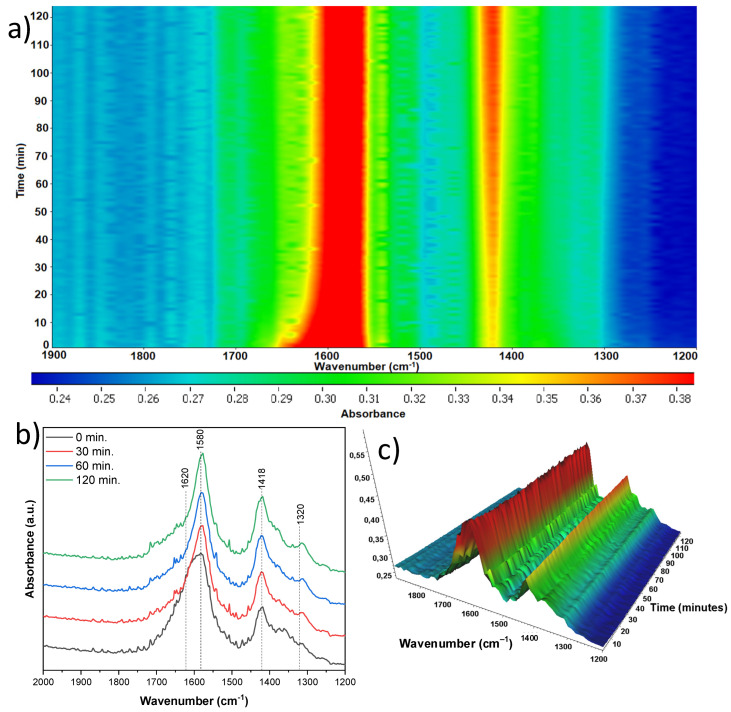
In situ FTIR of A-TNR-5 during the photocatalytic removal of acetaldehyde: (**a**) 2D mapping projection of the A = f(t, ν) relationship, (**b**) A = f(ν) graph after 0, 30, 60, and 120 min, (**c**) 3D mapping of the A = f(t, ν) relationship.

**Figure 17 materials-18-00986-f017:**
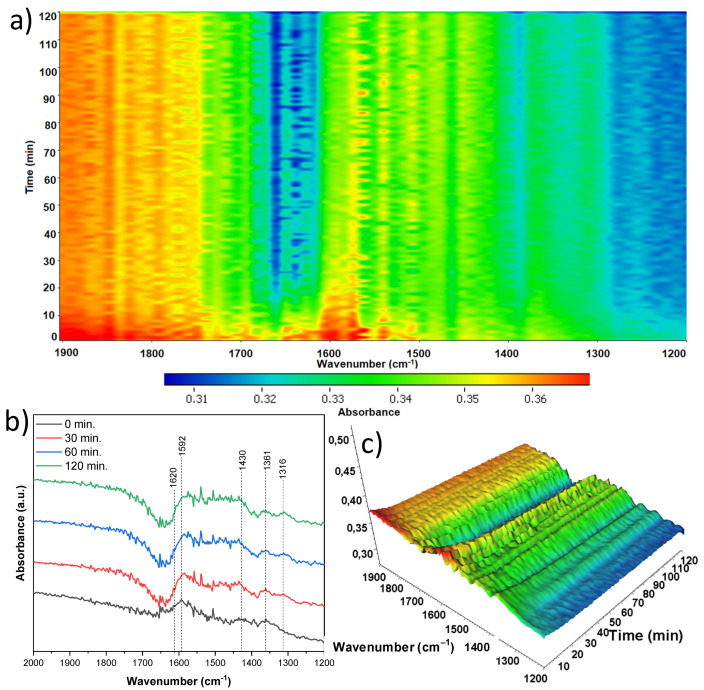
In situ FTIR of A-TNR-10 during the photocatalytic removal of acetaldehyde: (**a**) 2D mapping projection of the A = f(t, ν) relationship, (**b**) A = f(ν) graph after 0, 30, 60, and 120 min, (**c**) 3D mapping of the A = f(t, ν) relationship.

**Figure 18 materials-18-00986-f018:**
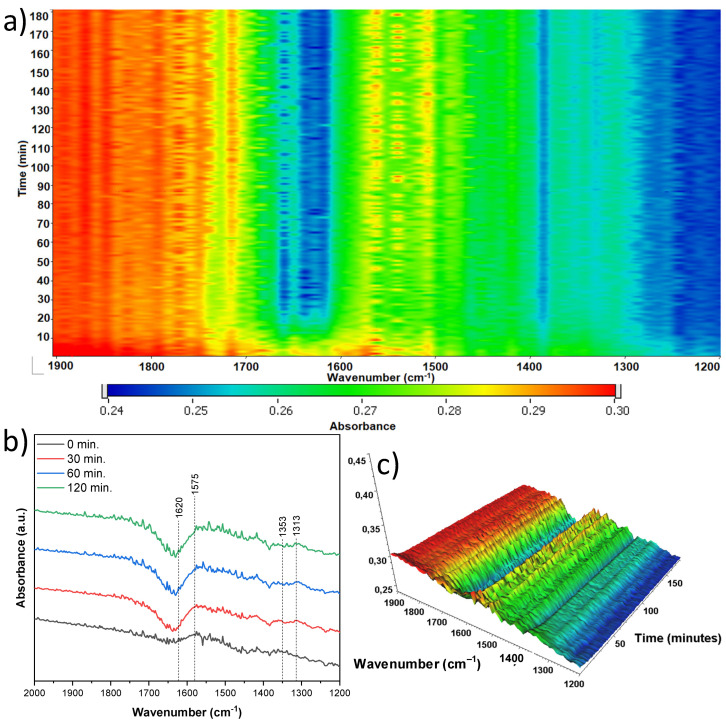
In situ FTIR of A-TNR-15 during the photocatalytic removal of acetaldehyde: (**a**) 2D mapping projection of the A = f(t, ν) relationship, (**b**) A = f(ν) graph after 0, 30, 60, and 120 min, (**c**) 3D mapping of the A = f(t, ν) relationship.

**Figure 19 materials-18-00986-f019:**
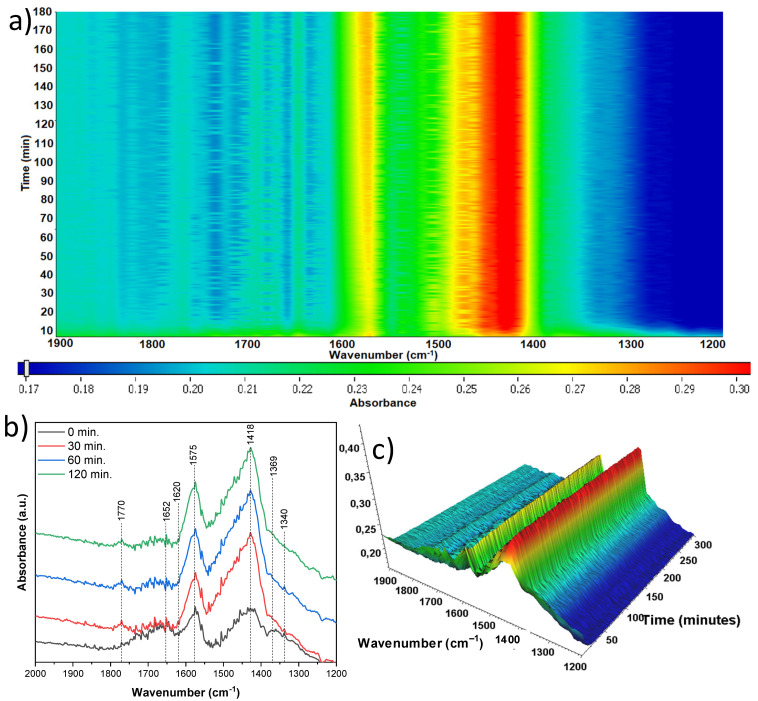
In situ FTIR of A/R-TNR-0 during the photocatalytic removal of acetaldehyde: (**a**) 2D mapping projection of the A = f(t, ν) relationship, (**b**) A = f(ν) graph after 0, 30, 60, and 120 min, (**c**) 3D mapping of the A = f(t, ν) relationship.

**Figure 20 materials-18-00986-f020:**
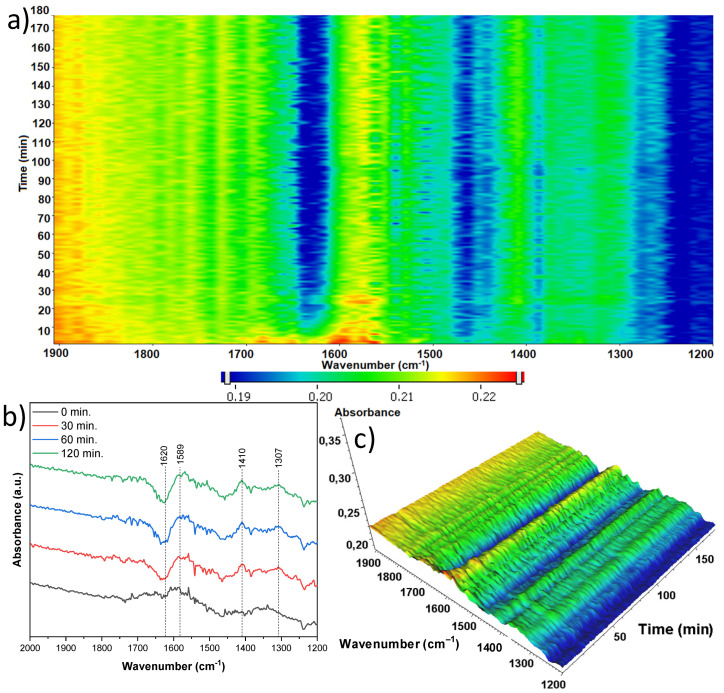
In situ FTIR of A/R-TNR-5 during the photocatalytic removal of acetaldehyde: (**a**) 2D mapping projection of the A = f(t, ν) relationship, (**b**) A = f(ν) graph after 0, 30, 60, and 120 min, (**c**) 3D mapping of the A = f(t, ν) relationship.

**Figure 21 materials-18-00986-f021:**
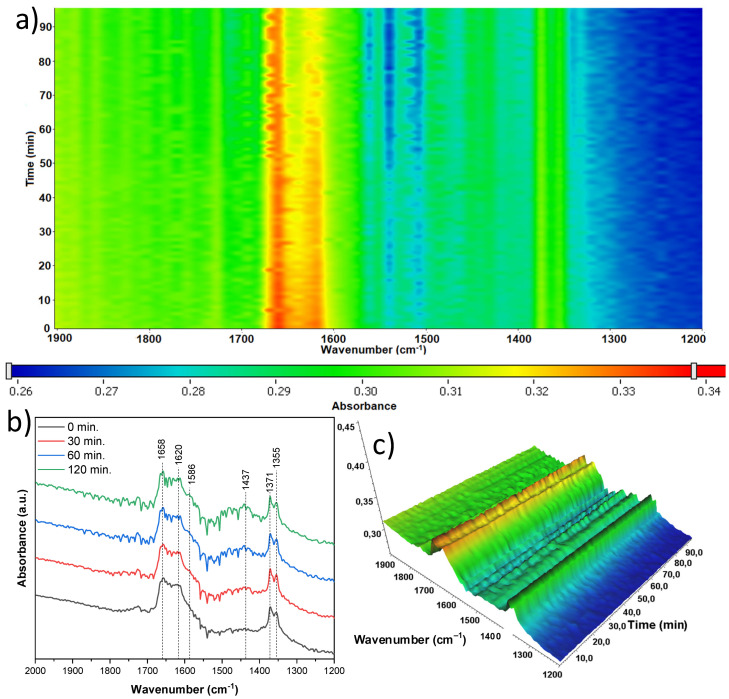
In situ FTIR of A/R-TNR-10 during the photocatalytic removal of acetaldehyde: (**a**) 2D mapping projection of the A = f(t, ν) relationship, (**b**) A = f(ν) graph after 0, 30, 60, and 120 min, (**c**) 3D mapping of the A = f(t, ν) relationship.

**Figure 22 materials-18-00986-f022:**
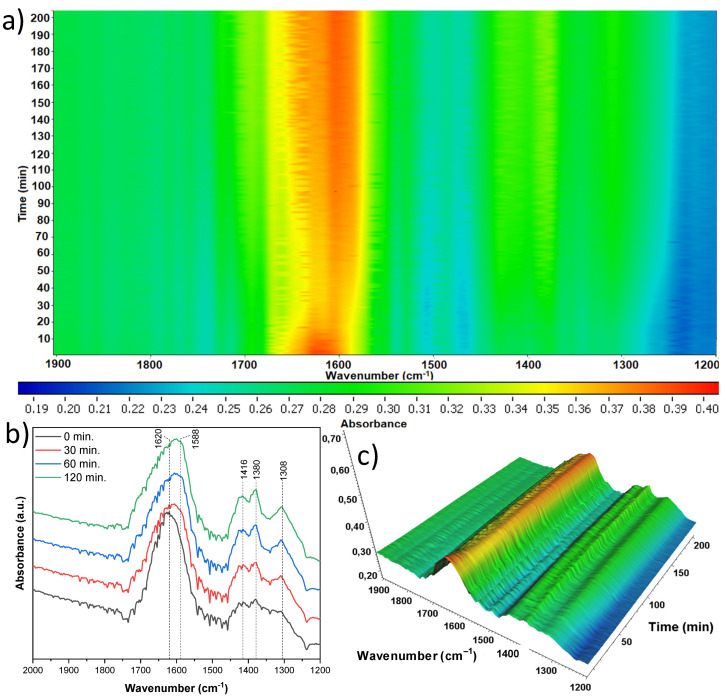
In situ FTIR of A/R-TNR-15 during the photocatalytic removal of acetaldehyde: (**a**) 2D mapping projection of the A = f(t, ν) relationship, (**b**) A = f(ν) graph after 0, 30, 60, and 120 min, (**c**) 3D mapping of the A = f(t, ν) relationship.

**Table 1 materials-18-00986-t001:** Selected properties of TiO_2_-based materials used for synthesis [26,27].

Sample	KRONOClean^®^7050	P25
Parameter		
Bulk density (g/dm^3^)	300	140
BET specific surface area (m^2^/g)	225	35–65
Phase composition (anatase/rutile)	100/0	85/15
Average crystallite size (nm)	15	20

**Table 2 materials-18-00986-t002:** XRF sodium content.

Sample	Na Content (%)
A/R-TNR-0	4.662
A/R-TNR-5	2.34
A/R-TNR-10	1.032
A/R-TNR-15	0.595
A-TNR-0	0
A-TNR-5	0
A-TNR-10	0
A-TNR-15	0

## Data Availability

All published data will be available in the open repository https://mostwiedzy.pl/ accessed on of 1 March 2025.

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
