# Peer review of "Preparation of TiO2 Nanorods@Ni-Foam for Photocatalytic Decomposition of Acetaldehyde—In Situ FTIR Surface Investigation"

_materials, 2025, doi:10.3390/ma18050986_

Round 1

Reviewer 1 Report

Comments and Suggestions for Authors

The study presents valuable insights into the preparation and photocatalytic performance of TNR@Ni-foam structures. A systematic in-situ FT-IR analysis was performed to understand the reaction pathway. This work could be considered for publication in Materials. However, several issues related to experimental methods, data analysis, results interpretation, and formatting need attention to enhance the clarity and accuracy of the manuscript.

1.      The preparation of TiO2 powder with 10 M NaOH at 150°C is mentioned, but specific details like reaction time and concentration control are missing. It is recommended to provide more details on the experimental conditions to help replicate the study.

2.      In the in-situ FTIR data, the spectra from different time points are presented together, making it difficult to track peak variations over time. It is suggested to separate the spectra for different time intervals to better observe the changes in infrared peaks.

3.      The mechanistic explanation of the catalytic performance is brief. It would be helpful to expand on how the TNR@Ni-foam structure enhances catalytic activity, potentially by discussing surface acidity, electronic structure, and comparisons to other similar systems.

4.      There are formatting inconsistencies in the chemical formulas, particularly with the placement of subscripts. Please review and ensure consistent formatting throughout the manuscript.

5.      The manuscript does not clarify why XRD analysis cannot be performed on Ni-supported TiO2. It is suggested to explain if Ni peaks interfere with TiO2 peaks or if protonation of TiO2 in the presence of excess HNO3 might affect the XRD results.

6.      The degradation efficiency of the system is not compared to other well-established catalysts. Including a comparison would help contextualize the results and demonstrate the competitive advantages or limitations of this material.

Reviewer 2 Report

Comments and Suggestions for Authors

Analytical Review Report

Comments to the authors:

The paper titled “Preparation of TiO2 nanorods@Ni-foam for photocatalytic de-composition of acetaldehyde – in situ FTIR surface investigation” aimed to investigate the acetaldehyde photodegradation during continuous flow of gas through a reactor coupled to FTIR. TNR@Ni-foam structures were prepared by non-organic, two-step strong base solution hydrothermal synthesis, while its morphology was observed by XRD and SEM.

After a detailed revision, since my point of view and considering the results presented by the authors, this work constitutes a nice addition to the mechanistic insight of the photodegradation of VOCs. However, certain minimal details have been identified throughout the manuscript that, if corrected, I believe could contribute to the understanding of the manuscript and improve it.

The following short report analyzes the paper's various aspects, highlighting its strengths and commendable qualities.

In general, the writing style is clear, concise, and engaging, with appropriate terminology and language. In addition, the presented paper has a good organization and logical structure, which contributes to the overall coherence and readability of the manuscript.

The results are in agreement with the study's objectives and address the research scope. The data are presented and supported by its analyses, contributing to the validity, clarity, and reliability of the findings. The XRD structure analysis clearly reflects the relationship between the morphology adopted by the material and the amount of Na remaining in it, which is verified by XRF.

The study presents robust experimental results supported by mechanistic FTIR considerations and supported by available bibliography. The discussion section provides concise analysis supported by literature. Based on experimental arguments, the conclusions are logical and sustain the ideas presented.

However, while the paper demonstrates these strengths, there are some areas where minor improvements could be made to enhance its quality further. These include:

Introduction:

-The introduction of the article lacks a review of the state of the art on the latest developments in in situ FTIR for mechanistic implications. It is recommended that the authors provide a short, up-to-date analysis of the relevant literature in the introduction. The journal´s reader should be able to understand the rationale for using this technique and its impact.

Experimental part:

-Page 3, line 109: “SEM images were taken using a SU 8200 Field Emission Scanning Electron Micro-109 scope with high resolution (Hitachi, Japan)” Please describe further specifications (kind of detector, etc.)

-Page 4, line 122: “...365 nm UV LED light source (LABIS, Poland).” Please describe the lamp specifications, such as configuration (diameter of optic fiber, potency of the lamp in Watts, as well as the irradiance intensity in mW/cm2/nm, or equivalent).

-Page 4, line 123: “The concentration of the outlet gas stream was measured by a gas chromatograph (SRI, USA) equipped with a flame ionization detector.” However, it is important to describe the specifications of the column and the temperature program used (if any).

Results and discussions:

Page 13, Fig. 13: Please improve the line thickness of the figure. The format in Fig. 15 can be used as a reference.

During the discussion of Fig. 13, the authors do not discuss the presence of reaction intermediates (for example, for A/R-TNR species). I consider it important to present the deconvolution of one of these graphs and to mention in the discussion the observation (or not) of intermediaries. Nor is there any mention of the presence of typical signals of “rotational movements” of the studied molecule and a possible reason for this. Is this acetaldehyde concentration optimal? What was the criterion used to decide this specific concentration? These are questions that can be addressed in an explanatory paragraph.

Fig. 15 to 22: For the reader’s understanding, please indicate at the bottom of the figure which graph belongs to one or other of the analyses described (a), (b) and (c).

Finally, I consider that the data reported in figures 13-22 make it particularly interesting to complement this study with a graph showing the time evolution of the FTIR spectra on the photocatalytic systems used (please refer to the study of Karelovic et al. Journal of Catalysis 427 (2023) 115119. Page 11; Figure 9 as an example).

This new graph would allow the description of the formation and consumption kinetics of each intermediate generated during the degradation of acetaldehyde, also providing key information on the stability of the intermediate products, possible reaction routes, and the efficiency of the photocatalytic process as a function of time.

In summary, incorporating these revisions will further strengthen the article, enhancing its clarity and impact. These improvements will not only reinforce the arguments presented but also increase the manuscript’s potential for acceptance, as it makes a significant contribution to advancing knowledge in the field of mechanistic insights into the photodegradation of model aldehydes.

Kind regards

Reviewer 3 Report

Comments and Suggestions for Authors

Dear authors,

 The study presents an interesting and well-executed approach to the preparation of TNR@Ni-foam composites by alkaline hydrothermal method, followed by acid leaching and a second hydrothermal treatment. The idea of ​​the paper is well-designed and interesting, and the research was conducted thoroughly and logically. Here are my comments:

Comment 1: English language corrections are required.

The section describing the synthesis is well-defined and presents a methodical approach for the preparation of Ni-foam/TiO2 nanorod composites via a two-step hydrothermal process. The use of nickel foam as a substrate for TiO2 deposition is promising, and the systematic variation of acid-washing conditions (with HNO3) offers valuable insight into synthesis optimization. The experiment was well performed, but there are a few points that could benefit from further clarification.

Comment 2: Line 88 - What effect, if any, does the angle at which the foam is placed in the autoclave (45 degrees) have on the final product? Would different placement angles (eg 0° or 90°) affect the uniformity of TiO2 deposition on the foam?

Comment 3: Line 95 - Can you elaborate on why the specific volumes of HNO3 (0, 5, 10 and 15 ml) were chosen for this step? How is this volume range determined?

Comment 4: Line 98 - Can you clarify how the temperature and treatment time were optimized for this step? Specifically, how did you choose 180°C, and is there evidence in the literature to support this particular temperature for the desired phase transformation?

Comment 5: Line 88 - What effect, if any, does the angle at which the foam is placed in the autoclave (45 degrees) have on the final product? Would different placement angles (eg 0° or 90°) affect the uniformity of TiO2 deposition on the foam?

Comment 6: Line 136 -  It can be seen that there is a change in the intensity of the diffraction peaks at different amounts of HNO₃. Is there a quantitative analysis of the phase composition (Rietveld refinement) that would confirm these claims?

Comment 6: Line 240 -  How do you explain the absence of sodium in A-TNR samples measured by the XRF technique? Is it possible that the amounts present are below the detection limit of the method?

Comment 7: I think the authors should add more relevant references with an emphasis on the last 5 years.

Comments on the Quality of English Language

This work has a lot of grammatical errors and I think the English needs to be improved.
